

# Development of a greenhouse-based inoculation protocol for the fungus *Colletotrichum cereale* pathogenic to annual bluegrass (*Poa annua*)

Lisa A. Beirn[1], Ruying Wang[1], Bruce B. Clarke[1] and Jo Anne Crouch[2]

[1] Department of Plant Biology & Pathology, Rutgers University, New Brunswick, NJ, USA
[2] Systematic Mycology & Microbiology, USDA-ARS, Beltsville, MD, USA

## ABSTRACT

The fungus *Colletotrichum cereale* incites anthracnose disease on *Poa annua* (annual bluegrass) turfgrass. Anthracnose disease is geographically widespread throughout the world and highly destructive to cool-season turfgrasses, with infections by *C. cereale* resulting in extensive turf loss. Comprehensive research aimed at controlling turfgrass anthracnose has been performed in the field, but knowledge of the causal organism and its basic biology is still needed. In particular, the lack of a reliable greenhouse-based inoculation protocol performed under controlled environmental conditions is an obstacle to the study of *C. cereale* and anthracnose disease. Our objective was to develop a consistent and reproducible inoculation protocol for the two major genetic lineages of *C. cereale*. By adapting previously successful field-based protocols and combining with components of existing inoculation procedures, the method we developed consistently produced *C. cereale* infection on two susceptible *P. annua* biotypes. Approximately 7 to 10 days post-inoculation, plants exhibited chlorosis and thinning consistent with anthracnose disease symptomology. Morphological inspection of inoculated plants revealed visual signs of the fungus (appressoria and acervuli), although acervuli were not always present. After stringent surface sterilization of inoculated host tissue, *C. cereale* was consistently re-isolated from symptomatic tissue. Real-time PCR detection analysis based on the *Apn2* marker confirmed the presence of the pathogen in host tissue, with both lineages of *C. cereale* detected from all inoculated plants. When a humidifier was not used, no infection developed for any biotypes or fungal isolates tested. The inoculation protocol described here marks significant progress for *in planta* studies of *C. cereale*, and will enable scientifically reproducible investigations of the biology, infectivity and lifestyle of this important grass pathogen.

## INTRODUCTION

Anthracnose, caused by the ascomycete fungus *Colletotrichum cereale* Manns *sensu lato* Crouch, Clarke & Hillman (2006), is a destructive disease of *Poa annua*

Corresponding author
Jo Anne Crouch,
JoAnne.Crouch@ars.usda.gov

L. f. reptans [Hauskins] T. Koyama (annual bluegrass) putting green turf. The disease has increased in frequency and severity over the past decade, leading to the development of environmentally sound best management practices that have reduced anthracnose severity on golf course putting greens in North America (*Murphy et al., 2008*). While improved anthracnose disease control measures are leading to reduced losses for many turfgrass management professionals (*Murphy et al., 2008*), essential knowledge about the biology and infectivity of this pathogen is still lacking. An understanding of virulence, host-pathogen interactions, the infection process, and pathogen genetics is vital to facilitate disease prediction, improve management practices, and enhance cultivar resistance. However, such advancements have been limited by the lack of a reliable, greenhouse-based experimental inoculation protocol, preventing the study of turfgrass anthracnose in a controlled and reproducible environment (*Murphy et al., 2008*).

Field inoculations of *P. annua* putting green turf with *C. cereale* are routinely performed (e.g., *Inguagiato, Murphy & Clarke, 2008*; *Inguagiato, Murphy & Clarke, 2009*). Briefly, the asexual spores (conidia) of *C. cereale* are harvested from axenic cultures, normalized to $5.0 \times 10^{-4}$ conidia $mL^{-1}$, and sprayed onto turf using a backpack sprayer approximately three hours before sundown during hot, humid weather when conditions are conductive for anthracnose disease (*Inguagiato, Murphy & Clarke, 2008*). This process is typically repeated for 3 to 5 successive days, with inoculated plants covered with polyethylene sheets overnight to prevent desiccation of conidia (*Inguagiato, Murphy & Clarke, 2008*). Early reports in the literature describe a similar procedure used for greenhouse inoculations of *P. annua* plants with spore solutions of *C. cereale* (*Bolton & Cordukes, 1981*; *Vargas & Detweiler, 1985*; *Vargas, Danneberger & Jones, 1993*), but consistent infection using these methods have been difficult to reproduce (*Murphy et al., 2008*). Inoculation of *P. annua* plants with agar plugs colonized by the pathogen have also produced inconsistent disease symptoms (*Backman, Landschoot & Huff, 1999*). Detached leaf assays were used to study the infection of *Agrostis stolonifera* L. (creeping bentgrass) grass plants by *C. cereale* (*Khan & Hsiang, 2003*). However, the use of detached leaf assays only provides information about the process of substrate utilization, and is unlikely to mirror the host-pathogen interaction that take place between *C. cereale* and intact grass plants. *Colletotrichum* pathogens of grasses utilize a hemibiotrophic strategy to colonize hosts, which is characterized by an initial stage of quiescent biotrophy, followed by a shift to necrotrophic colonization and degradation of host tissue, resulting in unique waves of differential gene expression taking place during each phase (*Crouch & Beirn, 2009*; *O'Connell et al., 2012*; *Crouch et al., 2014*). Gene expression studies of the hemibiotrophic infection strategy have revealed significantly different expression profiles in attached and detached Arabidopsis leaves when inoculated with *C. higginsianum* (*Liu et al., 2007*), thus detached leaf assays may not accurately reflect what is occurring *in planta*.

Natural populations of *C. cereale* are diverse, with ten distinct populations described from DNA sequence analysis using four markers (*Crouch et al., 2009b*). Moreover, the species has been subdivided into two primary genetic lineages, termed clade A and B (*Crouch, Clarke & Hillman, 2006*). Although divergent, these two major clades are

connected through gene flow sufficient to maintain the lineages as a single phylogenetic species (*Crouch et al., 2008*; *Crouch, Clarke & Hillman, 2009*; *Crouch et al., 2009b*). Among isolates of *C. cereale* collected from *P. annua* and *A. stolonifera* putting greens, the distribution of clades A and B is strongly influenced by geographic and host origin (*Beirn, Clarke & Crouch, 2014*), Clade A predominates in the southern US, regardless of host (*Beirn, Clarke & Crouch, 2014*). In contrast, clade A and clade B isolates have been found in equal frequencies and both lineages can occur on both hosts in the northern US; however, clade A isolates are found more frequently on *P. annua* and clade B is more common on *A. stolonifera* (*Beirn, Clarke & Crouch, 2014*). It is currently not known whether these population-scale differences affect infectivity or the ability of the pathogen to colonize different host species, but this association of specific genotypes in natural populations could explain the difficulties surrounding the development of a consistent, reproducible inoculation protocol for *C. cereale*.

The development of a reliable, greenhouse-based inoculation protocol for both *C. cereale* lineages would allow researchers to explore many unanswered questions about the disease cycle, host-pathogen interactions, and biology of *C. cereale* in a controlled environment. Therefore, the objective of this study was to develop a greenhouse-based inoculation protocol for both *C. cereale* genetic lineages by adapting and modifying existing spore solution inoculation methods and field-based protocols.

## MATERIALS AND METHODS

### Turfgrass hosts

*Poa annua* biotypes 98226, 99112 and 9712 were seeded (0.2 g per pot) in 10 cm pots filled with Fafard Canadian Grow Mix 2 (Agawam, MA). All biotypes were originally collected as seed from putting greens in the US. Biotypes were selected based on their susceptibility to anthracnose disease in turfgrass research plots at the Pennsylvania State University in State College, Pennsylvania (Dave Huff, pers. comm., 2011). Prior to inoculation, each biotype was tested for the presence of endophytic fungi. *Colletotrichum cereale* can survive asymptomatically in host tissue (*Crouch et al., 2009b*), thus the presence of endophytic *C. cereale* strains could confound real-time PCR results. In addition, each biotype was screened for existing mycorrhizal associations. For endophyte testing, 2 g of leaf tissue was randomly selected from 10 cm pots, trimmed to fit inside a 25 mL polypropylene tube (BD Falcon, Bedford, MA), then washed for 2 min in 70% EtOH, 2 min in 10% commercial NaClO, three rinses for 1 min in sterile $dH_2O$, and 10 s in 95% EtOH. Surface sterilized tissue was plated onto malt extract agar (MEA, Fisher Scientific, Pittsburgh, PA), and incubated at room temperature under continuous light for 3 weeks. For mycorrhizae testing, 0.5 g of *P. annua* roots were selected at random from established plants. Harvested roots were washed with $dH_2O$ and stained for mycorrhizae following a modified procedure of *Phillips & Haymanm (1970)*. Briefly, 1.5 cm root segments were heated in 10% KOH for 1 h at 90 °C. Cleared root segments were rinsed in tap water for 2 min, heated for 1 h in 20% HCl, followed by a final heating in 0.1% trypan blue (Roche Applied Science, Indianapolis, IN) for 30 min. Stained roots were mounted on microscope slides and

visualized using an Olympus BX41 clinical 71 microscope (New York/New Jersey Scientific, Middlebush, NJ).

## Fungal isolates

Two isolates of *C. cereale* were used in this study. Isolate TCGC5-63 (clade A) was collected from *P. annua* in 2002 from a golf course putting green in Temecula, CA, whereas isolate HF217CS (clade B) was collected from *P. annua* in 2012 from turfgrass research plots in North Brunswick, NJ. Fungal specimens are deposited in Centraalbureau voor Schimmelcultures (Utrecht, The Netherlands). Prior to inoculation, isolates were removed from −80 °C storage, where they were maintained as desiccated conidia on silica gel in 1 mL cryogenic tubes (Fisher Scientific, Pittsburgh, PA), and placed on potato dextrose agar (PDA, Fisher Scientific, Pittsburgh, PA) under continuous light at room temperature. After fungal growth covered the PDA plates (∼8–10 days), 10 mL of sterile dH$_2$O was poured onto the plate and fungal mycelia and spores were gently removed by scraping with a sterile glass rod. A sterile inoculation loop was used to streak the fungal suspension onto a fresh PDA plate. Streaked plates were not sealed with parafilm and were placed under direct light at room temperature to encourage sporulation (∼3–4 days).

## Inoculation protocol

All *P. annua* biotypes were inoculated with each *C. cereale* isolate in separate treatments, with three replications per biotype. The inoculation protocol consisted of a 20 mL conidial suspension ($10^{-6}$ conidia/mL) of *C. cereale* and 10% potato dextrose broth (PDB, Fisher Scientific, Pittsburgh, PA). The conidial suspension was sprayed onto the foliage of 8 wk post-emergent *P. annua* plants using a hand atomizer. Non-inoculated plants were sprayed with 20 mL 10% PDB as a control. Following inoculation, plants were placed in a custom-made plexiglass mist chamber inside a growth chamber for 24 h. The plexiglass chamber measured 91.5 cm × 63.5 cm × 76.2 cm and was constructed with 7.9 mm plexiglass. A humidifier (Herrmidifier, Phoenix, AZ) atomized water into water vapor, providing a source of moisture in the chamber. The humidifier ran for a duration of 1 h every 2 h. Following incubation, plants were removed from the plexiglass chamber and remained in the growth chamber under conditions of 12 h daylight (500 µE m$^{-2}$ s$^{-1}$), 80% RH, 30 °C day, and 26 °C night until symptom development (∼7 to 10 days). All plants were watered every other day at the base to avoid wetting inoculated foliage during the course of the experiments. The experiment was replicated twice under the same conditions described above and included negative controls. To test the effects of moisture and plant growth medium on the ability of *C. cereale* to infect *P. annua*, the experiment was also repeated under the same growth chamber conditions two additional times, with variations in moisture and plant growth substrate. In one experiment, inoculated plants were seeded in Grow Mix 2, but no humidifier was used during the inoculation period. In the other experiment, plants were seeded in sand and inoculated and moisture was supplied as previously described.

## Confirmation of infection

Infection was confirmed in three ways: (1) inoculated plants were visually inspected for signs of the fungus (appressoria and acervuli), (2) *C. cereale* was re-isolated from inoculated tissue after symptom development, and (3) inoculated plants were tested for the presence of *C. cereale* using real-time PCR, clade-specific assays developed using the *Apn2* marker (*Beirn, Clarke & Crouch, 2014*). This assay confirms to the Minimum Information for Publication of Quantitative Real-Time PCR Experiments guidelines; complete details describing the assay can be found in *Beirn, Clarke & Crouch (2014)*. For pathogen isolations and real-time PCR, plant tissue was surfaced sterilized prior to plating or DNA analysis with 10% commercial NaClO for 2 min, 70% EtOH for 2 min, rinsed in sterile dH$_2$O, and allowed to dry in a sterile hood. Half of the dried tissue was plated onto PDA, while the remaining tissue was used for real-time PCR analysis.

DNA was extracted from the remaining tissue using the OmniPrep DNA Extraction Kit (G-BioSciences, St. Louis, MO) and a modified manufacturer's protocol. Briefly, 0.5 g of plant tissue was placed in a 2 mL microcentrifuge tube with eight 2.5 mm glass beads (BioSpec Products, Barletsville, OK) and beat in a BioSpec bead-beater (Barletsville, OK) on the medium setting for six minutes. The manufacturer's protocol was followed for the remaining steps. Real-time PCR reactions were performed as previously described (*Beirn, Clarke & Crouch, 2014*) using a Cepheid SmartCycler (Cepheid, Sunnyvale, CA) in 25 μl Cepheid tubes. Briefly, Cepheid's Smartmix HM lyophilized PCR master mix was used for all reactions with a final probe concentration of 2 μM and primer concentration of 20 μM. Cycling conditions were as follows: initial denaturation at 95 °C for 120 s, followed by 45 cycles of 95 °C for 5 s, 60 °C anneal for 30 s, and 72 °C extension for 1 s. Samples were considered positive if the fluorescent threshold (30) was crossed prior to cycle 40 and negative if samples produced cycle threshold ($C_T$) values equal to zero.

# RESULTS

## Mycorrhizae and endophyte testing

No visual signs of endophytic fungi or mycorrhizae were found in any of the *P. annua* biotypes used in this study.

## Inoculation results

Appressoria formation was observed on *P. annua* biotypes 99112 and 98226 ∼24 h after inoculation with both *C. cereale* isolates on plants grown in potting mix and sand (Fig. 1A). Appressoria were randomly distributed on leaf surfaces and were dark brown/black and rounded or lobed, measuring 9–10 μm by 6–9.5 μm. Disease symptoms were not observed until 6 to 9 days after appressoria formation on *P. annua* biotypes 99112 and 98226, and were similar for plants seeded in potting mix or sand. Although leaf lesions were never present, plants inoculated with *C. cereale* isolate TCGC5-63 or isolate HF217CS appeared chlorotic when compared to uninoculated controls (Fig. 1B). Acervuli with diagnostic black setae were only present on plants inoculated with *C. cereale* isolate TCGC5-63

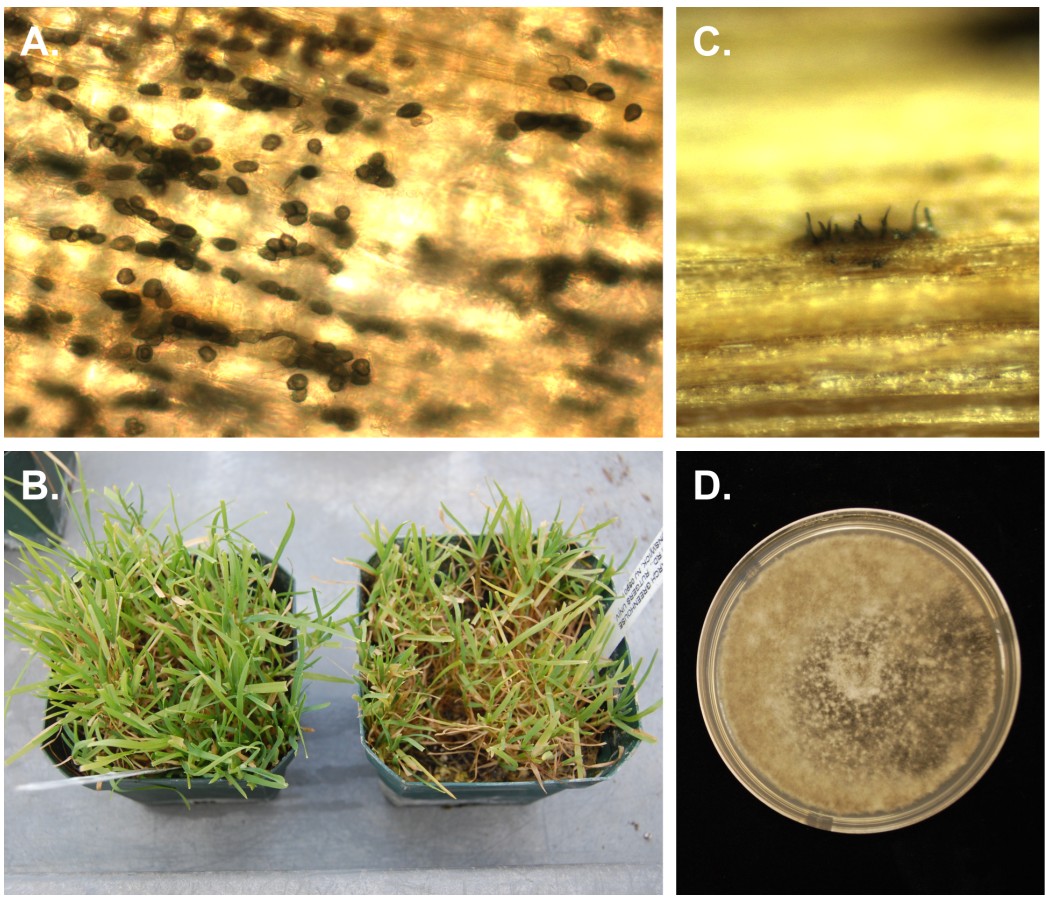

**Figure 1** **Photographs of *Poa annua* plants inoculated with *Colletotrichum cereale*.** (A) Numerous appressoria of *C. cereale* on inoculated leaf, (B) *P. annua* biotype 98226 uninoculated control on the left, compared to plants of the same biotype inoculated with *C. cereale* on the right, (C) *C. cereale* isolate TCGC5-63 acervulus on inoculated plant tissue, and (D) *C. cereale* culture TCGC5-63 re-isolated from inoculated tissue.

(Fig. 1C). Numerous falcate, hyaline conidia were found in conjunction with acervuli. *P. annua* biotype 9712 never developed any leaf symptoms or signs and was negative for all tests for the presence of *C. cereale*. Negative controls did not develop any disease symptoms, and did not exhibit any signs of fungal colonization.

Both isolates of *C. cereale* were re-isolated from symptomatic plant tissues on PDA. Mycelium was white at first, then turning slightly gray with age (Fig. 1D). Numerous acervuli were formed on the agar surface, forming in the older parts of the colony first. Conidia appeared orange colored in mass and measurements conformed to those reported for *C. cereale* (*Crouch, Clarke & Hillman, 2006*). Cultures of *C. cereale* were recovered from all symptomatic plants, regardless of whether they were seeded in potting mix or sand. Control plants generated no isolates of *C. cereale* upon culturing.

No infection was present in inoculated plants where the humidifier was not used.

### Real-time PCR confirmation

Leaf tissue samples were taken from all plants to either confirm infection or, in the case of the controls, to verify the absence of infection. Real-time PCR confirmed infection for both *C. cereale* isolates HF217CS and TCGC5-63 on *P. annua* biotypes 99112 and 98226, with average $C_T$ values of 32.90 and 32.04, respectively. Control plants and inoculated biotype 9712 generated $C_T$ values of 0.00. DNA sequences of the ITS region amplified from the isolated fungi matched *C. cereale* isolates in the GenBank database with 100% similarity (data not shown).

## DISCUSSION

The primary objective of this study was to develop a repeatable inoculation protocol for the two *C. cereale* genetic lineages commonly found on *P. annua*. Two biotypes of *P. annua* were successfully inoculated with a clade A and clade B isolate of *C. cereale*, representing the first documented occurrence of a successful inoculation protocol for both genetic lineages of *C. cereale* on the same plant biotype. The methods developed here are similar to methods reported in earlier literature (e.g., *Bolton & Cordukes, 1981*; *Vargas & Detweiler, 1985*; *Vargas, Danneberger & Jones, 1993*), but, unlike these studies, lesions were never observed on inoculated plant material in our study. However, inoculated plants did appear visibly chlorotic and thinned, *C. cereale* was successfully isolated from symptomatic hosts, and real-time PCR detected the presence of the fungus after rigorous surface sterilization. These findings reflect what was recently observed in the centipedegrass-anthracnose pathosystem, where inoculated plants appeared chlorotic and, over time, necrotic, with no acervuli present (*Crouch & Tomaso-Peterson, 2012*). Molecular detection methods, in combination with Koch's postulates, also confirmed pathogenicity in this system (*Crouch & Tomaso-Peterson, 2012*).

One *P. annua* biotype, 9712, never developed infection with *C. cereale*, despite displaying susceptibility in the field. Early research with *C. cereale* suggested the presence of multiple races in cereal plants, as certain strains of the fungus were not able to infect all cultivars of cereal crops tested (*Sanford, 1935*; *Bruehl, 1948*; *Bell, 1949*), though race genotyping has never been performed. Cultivar-race specificity is known to occur in *C. lindemuthianum*, the causal agent of bean anthracnose (*Barrus, 1911*), thus the presence of multiple races in *C. cereale* would not be surprising. Though additional data is required to test this hypothesis, this observation may explain why *C. cereale* inoculations have been difficult to reproduce in previous field and greenhouse studies. In addition, the lack of acervuli produced *in planta* in many studies may also be a contributing factor; some isolates used in inoculations may have little affinity for forming acervuli *in planta*, making it difficult to verify the presence of the fungus. Without molecular technologies and this diagnostic feature, confirming the presence of *C. cereale* is dependent on re-isolating the fungus, a task that can be difficult and time consuming. Our real-time PCR probes (*Beirn, Clarke & Crouch, 2014*) provide a reliable method for testing the presence of *C. cereale in planta*, allowing for quick disease diagnosis, pathogen quantification, and lineage genotyping.

Our data shows that moisture on the leaf surface over an extended period is required for establishing *C. cereale* infection in a controlled setting. This supports the observations of *Vargas, Danneberger & Jones (1993)*, where increasing leaf wetness from 12 to 72 h drastically increased the percent of *C. cereale* infections on inoculated plants. Extended periods of leaf wetness also appears to be a requirement for other grass infecting *Colletotrichum* species. Inoculation protocols developed for *C. eremochloae*, the causal agent of centipedegrass anthracnose, and *C. navitas*, the causal agent of anthracnose disease of switchgrass, both describe placing plants in plastic containers or bags following spraying with conidial solutions (*Crouch & Tomaso-Peterson, 2012*; *Crouch et al., 2009a*). This practice not only increases humidity, but also encourages water droplet formation within the container, many of which often drip back to the leaf surface. Without excess moisture, conidia may face desiccation before they are able to germinate, thereby preventing infection. While we did not evaluate whether bagging plants inoculated with *C. cereale* results in infection similar to that obtained with using the humidity chamber, the success of this technique with other grass-infecting *Colletotrichum* species suggests that this may be a viable option for those seeking to study *C. cereale* infection when a humidity chamber is not available.

The development of a rapid and reliable, greenhouse-based inoculation protocol marks a significant milestone for advancing *in planta* studies of *C. cereale*. This tool will serve as a foundation for investigating the biology and infectivity of this important pathogen under controlled environmental conditions. Further, specific questions about the infectivity of both *C. cereale* lineages as well as rapid screening of turfgrass germplasm to speed the development of cultivars with resistance to anthracnose can now be addressed.

## ACKNOWLEDGEMENTS

We thank Dave Huff for supplying seed of *P. annua* biotypes, Mark Peacos for support building the misting chamber, and Joe Florentine and Jeff Akers for assistance with the growth chamber.

### Funding

This research was funded by the Rutgers Center for Turfgrass Science, the United States Department of Agriculture (USDA), Agricultural Research Service and the USDA National Institute of Food and Agriculture. Mention of trade names or commercial products in this publication is solely for the purpose of providing specific information and does not imply recommendation or endorsement by the USDA. The funders had no role in study design, data collection and analysis, decision to publish, or preparation of the manuscript.

### Grant Disclosures

The following grant information was disclosed by the authors:
Rutgers Center for Turfgrass Science.
The United States Department of Agriculture (USDA).

Agricultural Research Service.

The USDA National Institute of Food and Agriculture.

## Competing Interests

The authors declare there are no competing interests.

## Author Contributions

- Lisa A. Beirn and Ruying Wang conceived and designed the experiments, performed the experiments, analyzed the data, wrote the paper, prepared figures and/or tables, reviewed drafts of the paper.
- Bruce B. Clarke and Jo Anne Crouch conceived and designed the experiments, contributed reagents/materials/analysis tools, wrote the paper, reviewed drafts of the paper.

## Data Availability

The following information was supplied regarding the deposition of related data:

All data is presented in the results section.

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
