# Peer review of "Development of a greenhouse-based inoculation protocol for the fungus Colletotrichum cereale pathogenic to annual bluegrass (Poa annua)"

_PeerJ, doi:10.7717/peerj.1153_

## Round 0.1 · original submission · Minor Revisions

The manuscript 'Development of a greenhouse-based inoculation protocol for the fungus Colletotrichum cereale pathogenic to annual bluegrass (Poa annua)' has been reviewed by 3 colleagues and the decision is for Minor Revisions.

Reviewer 1 ·

Basic reporting

The paper follows PeerJ policies, it is clearly written, cite relevant literature mainly from the same group but omit to highlight what are the crucial differences of previous infection methods used with C. species compared to this one.
qPCR figure is not needed or should be modified to increase the image quality and with better legend for its reading.
The paper seems to be part of a current research looking at races. The method description, given its limited novelty, would better fit in a research paper describing pathogenic diversity within the species.

Experimental design

The paper respects Koch’s postulate but some further data on the reproducibility of symptoms should be provided given the claim of the title. To show that infection is reproducible would require observation and scoring of a set of biological replicates done in different time with some kind of statistical analysis associated (like repeat the experiment in different time of the year and check on symptoms reproducibility showing quantitative scales for disease assessment).
According to Peer J policy “all quantitative real time PCR should follow the MIQE guidelines (the Minimum Information for Publication of Quantitative Real-Time PCR Experiments) and checklist”. Please provide evidence that this was the case in this protocol description.

Validity of the findings

The paper is not a complete research work as it describes a method for inoculation showing that 2 isolates behave differently in the relationship with bluegrass biotypes (not really enough to do a complete study on the existence of races in the species). At the same time it is not a new method paper apart from the species used (a very banal transfer of an established technique to a new pathosystem, see for example Avila-Adame et al 2003 PLANT DISEASE)
It lacks statistical evaluation of symptoms reproducibility.

Additional comments

The paper by Beirn et al describe the protocol of infection of C.cereale on annual bluegrass.
The paper is mainly methodological description of the inoculation protocol. Authors claim that for the first time they achieved a greenhouse reproducible infection of C. cereale on annual bluegrass. To prove it, they used 3 plant biotypes and 2 C. cereale isolates showing that 2 of the 3 biotypes were successfully infected in greenhouse testing
Originality of the research is only partially accomplished as the suggested hypothesis of the presence of races within the pathogen is not further investigated. Therefore the paper should be only judged on the verification that a protocol similarly established for other Colletothrichum species is also applicable to the bluegrass-C.cereale pathosystem. If only the method should be evaluated the paper fails to be a fully complete methodological paper as no statistics is carried out on symptoms and their reproducibility with many biological replicates carrried out in different times.

I would invite authors to generate a complete piece of research by adding information and research on the possible presence of races. Otherwise the originality of the paper is quite limited.

Reviewer 2 ·

Basic reporting

No comment, the article meets standards required by the journal.

Experimental design

Experimental design is appropriate. A better description of how the plants were treated is required. A humidifier was used but the authors use the terminology "mist".

Validity of the findings

Findings are within the scope expected with the exception that one cultivar did not develop disease. The authors did address this in the discussion but their technique will not work for every plant/isolate combination.

Additional comments

Comments have been made on the .pdf. The use of "misting" or "misting chamber" is misleading if plants were not treated with mist which is different than placing plants in a water vapor saturated environment. I do not like the use of "lifestyle" when trying to describe ecology.

Annotated reviews are not available for download in order to protect the identity of reviewers who chose to remain anonymous.

·

Basic reporting

Relevant, well written and well structured

Experimental design

Clearly deffined experimental design, according to objective

Validity of the findings

Results clearly represent an advance on the field, although they also raise new questions, clearly marked as speculative, on the possibility of existance of races in this pathosystem

Additional comments

L102 space missing after H2O
L105 replace ‘2 m’ by ‘2 min’
L106 replace ‘30 m’ by ‘30 min’
L122-123 please revise concentration (10^6 conidia/mL?)
L145 replace ‘2 m’ by ‘2 min’ (twice)
L145 10% NaClO? Do you mean 10% commercial bleach? Typically commercial bleach is 5% NaClO, so 10% NaClO would mean ‘double strength bleach’. Two minutes in ‘double strength bleach’ would probably destroy the tissues.
L180 No information is given about plants seeded in sand besides that on appressoria (L165)
L313 Shouldn’t there be an ‘et al.’ here?

---

## Round 0.2 · accepted · Accept

The revised version has been made taking into account the comments of all the reviewers. Therefore, the manuscript can be accepted in PeerJ.